# GUI-Rise: Structured Reasoning and History Summarization for GUI Navigation

**Tao Liu**[1*]  **Chongyu Wang**[1*]  **Rongjie Li**[2]

**Yingchen Yu**[2]  **Xuming He**[1,3†]  **Bai Song**[2]

[1]ShanghaiTech University  [2]ByteDance
[3]Shanghai Engineering Research Center of Intelligent Vision and Imaging
`{liutao, wangchy, hexm}@shanghaitech.edu.cn`
`lirj@bytedance.com, yingchen001@e.ntu.edu.sg, songbai.site@gmail.com`

## Abstract

While Multimodal Large Language Models (MLLMS) have advanced GUI navigation agents, current approaches face limitations in cross-domain generalization and effective history utilization. We present a reasoning-enhanced framework that systematically integrates structured reasoning, action prediction, and history summarization. The structured reasoning component generates coherent Chain-of-Thought analyses combining progress estimation and decision reasoning, which inform both immediate action predictions and compact history summaries for future steps. Based on this framework, we train a GUI agent, **GUI-Rise**, through supervised fine-tuning on pseudo-labeled trajectories and reinforcement learning with Group Relative Policy Optimization (GRPO). This framework employs specialized rewards, including a history-aware objective, directly linking summary quality to subsequent action performance. Comprehensive evaluations on standard benchmarks demonstrate state-of-the-art results under identical training data conditions, with particularly strong performance in out-of-domain scenarios. These findings validate our framework's ability to maintain robust reasoning and generalization across diverse GUI navigation tasks. Code is available at `https://leon022.github.io/GUI-Rise`.

## 1 Introduction

Recent advances in multimodal artificial intelligence [4, 48, 1, 12] have reignited interest in agents that can autonomously navigate Graphical User Interfaces (GUIs). By translating natural-language instructions into actions on screen, these agents hold the promise of reshaping human–computer interaction and streamlining everyday workflows [8, 19, 23, 46, 37]. At the core of this progress are Multimodal Large Language Models (MLLMs) [18, 17, 7, 3], which combine visual perception with linguistic reasoning to identify, interpret, and manipulate interface elements [14, 49]. Despite this promise, deploying GUI agents in real-world applications remains challenging: an effective agent must sustain coherent behaviour over multi-step interactions, continuously reason about the evolving interface state, and integrate its own history.

Although notable progress has been made in multi-step GUI navigation [40, 49, 6, 37, 19], current agents remain far from reliable. Approaches that lean on prompt engineering with proprietary LLMs such as GPT-4 [1, 46, 52] are constrained by the frozen capabilities of the underlying model and cannot be easily adapted to new domains. Supervised Fine-Tuning (SFT) on open-source backbones

---

[*]These authors contributed equally to this work.

[†]Corresponding author

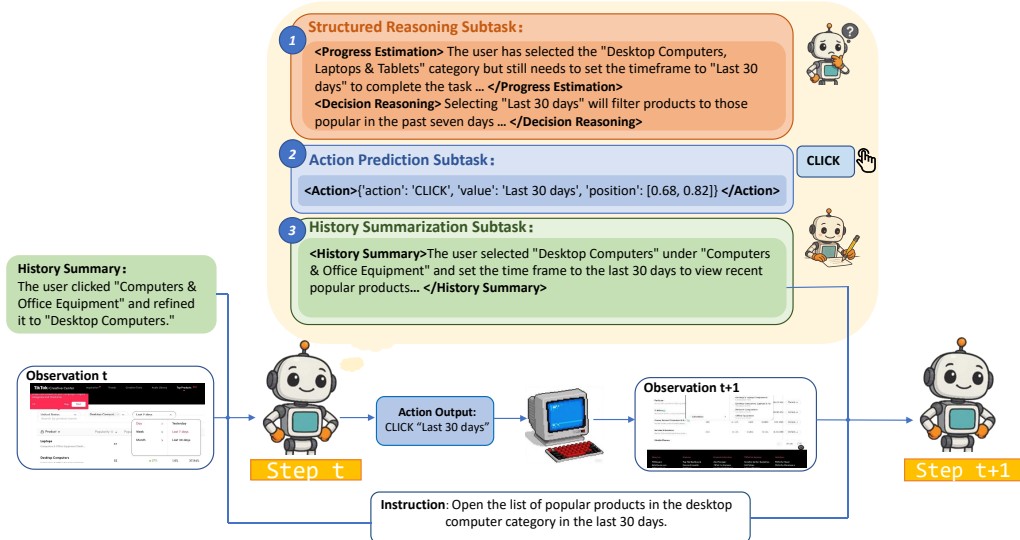

Figure 1: GUI-Rise agent framework overview. It introduces a three-subtask framework that integrates structured reasoning, action prediction, and history summarization. At each step, the agent performs structured reasoning (progress estimation and decision analysis), predicts the next GUI action, and updates a compact history summary for the next iteration.

[8, 19, 23] improves in-distribution accuracy, but often overfits to static instruction–action pairs and fails to generalize. A deeper obstacle is the need for long-horizon, sequential reasoning [24, 49, 27]: effective decisions must depend on what the agent has already done and how the interface has evolved. Existing systems encode history either (i) as action sequences alone [8, 9], which omit visual state and hinder progress estimation, or (ii) as full-screen screenshots [19, 23, 29, 50], which are computationally expensive and force severely truncated context windows. Consequently, unlike humans—who effortlessly integrate past observations and estimate task progress [14]—today's GUI agents still struggle with coherent, long-term reasoning.

To address these limitations, we propose a reasoning-enhanced GUI navigation framework that couples concise history summarization with explicit, structured reasoning. Based this framework, we train a GUI agent, GUI-Rise. GUI-Rise operates in a step-by-step cycle: it compresses the full interaction trace into a short textual summary, performs structured chain-of-thought reasoning, and predicts the next GUI action, as illustrated in Figure 1. This design imitates the way humans navigate interfaces, allowing the agent to maintain coherence and goal awareness over long action sequences. To achieve this capability, we introduce a reinforcement learning-based training strategy where the agent interacts with a simulated environment to develop adaptive, structured reasoning skills tailored to GUI tasks.

Specifically, GUI-Rise operates through three core subtasks at each interaction step: structured reasoning, action prediction, and history summarization. The agent first analyzes the current screen observation alongside previous interactions to assess task progress, then predicts the optimal next action based on this analysis, and finally updates its interaction history to maintain an evolving context for future decisions. To enable this structured reasoning capability, we develop a two-stage training paradigm. The first stage bootstraps the model on a small, synthetically labeled dataset to establish basic reasoning and summarization skills, while the second stage adopts reinforcement learning in a simulated GUI environment to refine task-specific reasoning strategies through interaction. Importantly, we employ Group Relative Policy Optimization (GRPO) with three complementary reward functions: 1) an action accuracy reward that evaluates prediction correctness, 2) a format reward that enforces structured reasoning through semantic tagging of components, and 3) a history summary reward that assesses the summary's quality for future decisions. This triad of rewards ensures the agent develops both precise action selection and robust reasoning capabilities while maintaining coherent, context-aware behavior throughout extended interactions.

We conduct extensive experiments on benchmarks including Mind2Web [9], AITW [30] and Mini-Wob [35], evaluating out-of-domain generalization in both offline and online settings. Our findings reveal that our proposed method yields substantial improvements in out-of-domain generalization under the same training data. In summary, our contributions are three-fold,

- We propose a reasoning-enhanced agent framework that formally structures the reasoning and execution pipeline for GUI-based tasks, introducing a novel history representation mechanism to improve decision accuracy.

- We develop a comprehensive reward combining action incentives with a specialized history summary reward, jointly optimizing for coherent reasoning and task-progressive outputs.

- Extensive experiments demonstrate our model's state-of-the-art performance across multiple benchmarks, with particularly strong results in out-of-domain scenarios, highlighting its superior generalization capabilities for real-world GUI navigation tasks.

## 2 Related Work

**GUI Agent.** In recent years, the rapid advancement of generative artificial intelligence [1, 12, 32] has made GUI agents a prominent research focus. Early studies [15, 27, 46, 49, 52] primarily focused on the design of application frameworks, relying heavily on the reasoning capabilities of close-source large language models (LLMs) [1], which often necessitated significant human intervention during practical deployment. Subsequent researches [19, 44, 42] began to explore the use of open-source LLMs [38, 3, 11] for GUI navigation, typically by collecting interaction data and training task-specific agent models. Some approaches [39, 22] further leveraged HTML document corpora, employing LLMs to perform structural parsing and long-context reasoning, thereby enabling autonomous interaction with GUI environments. Recent works such as UI-TARs [29] leverage large-scale reasoning data, including GUI screenshots, to boost performance through task-specific fine-tuning, though this also poses challenges for generalizing reasoning to unseen domains. To address this challenge, our work introduces a structured reasoning Chain-of-Thought (CoT) that mimics human-like decision-making through interpretable steps such as progress estimation and decision reasoning, and further enhance generalization via reinforcement learning with tailored rewards.

**Memory in GUI Agent.** The memory mechanism [14] of intelligent agents is crucial in GUI tasks, storing necessary historical information. It's mainly divided into short-term memory (STM) and long-term memory (LTM) based on storage duration and functionality. LTM [36, 16, 47, 41] is a persistent knowledge storage. It records global information like trajectories, environmental states, and user preferences from historical interactions, highlighting the importance of LTM in complex scenarios. STM focuses on the agent's immediate context for logical consistency during task execution. Early works like the action-only series [46, 16, 8, 9] used action sequence recording for task state perception. Later, studies such as ShowUI [19, 29, 40, 23] combined "actions and screenshots" within a fixed-length window for better history modeling. More recently, UI-Hawk [50] reduces visual history images to one-quarter of their original size and adopts a visual token compression ratio of 16, improving computational efficiency and inference performance. However, low-resolution images may lose fine-grained details, such as the states of small interactive buttons, compromising the integrity of historical information. Consequently, both forms of short-term memory (STM) still suffer from historical information loss, leading to inaccurate decision-making. Our approach integrates history summarization into agent reasoning, enabling efficient execution history representation without fixed window limits or external semantic processing, and improving memory efficiency and adaptability to complex tasks.

**Reinforcement Learning for LLMs/MLLMs.** Reinforcement Learning (RL) has emerged as a powerful tool for enhancing the capabilities of LLMs[1] and MLLMs[3]. While Proximal Policy Optimization (PPO) [31] is widely adopted, its reliance on value networks incurs high computational cost and instability. To overcome this, GRPO [33] replaces value estimation with inter-group comparisons, enabling more efficient and stable training, and has shown success in code generation [20, 43] and mathematical reasoning [12, 10]. Recent works [5, 34, 53] have further extended this algorithm to multimodal tasks such as visual counting and grounding. Among them, UI-R1 [25] applies GRPO to GUI navigation but is limited to single-step tasks. In contrast, we explore the more complex

multi-step GUI navigation setting, introducing a history summarization objective trained with RL to support long-horizon reasoning and better generalization.

## 3 Method

We present a reasoning-enhanced framework for GUI navigation that formalizes the interaction process through three core components: (1) structured reasoning, (2) action prediction, and (3) history summarization. Our MLLM-based agent, GUI-Rise, processes multimodal inputs—including the current screen observation, user instruction, and interaction history—to generate coherent outputs comprising a CoT analysis, executable action, and updated context summary (Figure 1). The remainder of this section details our approach: Section 3.1 formalizes the GUI navigation problem, Section 3.2 presents the agent's architecture, and Section 3.3 elaborates on each subtask's design and integration.

### 3.1 Task Definition

We define a general GUI navigation agent that executes natural language instructions $\mathbf{u}$ by performing a sequence of GUI-level actions. At each time step $t$, the agent $\pi$, parameterized by $\theta$, observes the environment state $\mathbf{s}_t = [\mathbf{o}_t, \mathbf{h}_{t-1}]$, and selects an action $\alpha_t$ based on its policy:

$$\alpha_t = \pi_\theta(\mathbf{u}, \mathbf{s}_t) \tag{1}$$

Here, $\mathbf{o}_t \in \mathbb{D}^{w \times h \times 3}$ denotes the visual observation of the GUI (e.g., a screenshot of resolution $w \times h$), and $\mathbf{h}_{t-1} \in \mathbb{D}^L$ represents the interaction history up to step $t-1$. The agent outputs atomic GUI-level actions $\alpha_t$, such as clicking a button or entering text. A formal definition of the action space is provided in Section 3.3. After executing $\alpha_t$, the environment transitions to $\mathbf{s}_{t+1}$, and the agent receives a scalar reward $r_t$:

$$r_t = \mathcal{R}(\mathbf{s}_t, \alpha_t, \mathbf{s}_{t+1}) \tag{2}$$

The reward function $\mathcal{R}$ encodes both task-specific progress and adherence to behavioral constraints, guiding the agent toward successful task completion.

### 3.2 Agent Architecture

GUI-Rise jointly processes visual inputs (GUI screenshots) and textual inputs (user instructions and interaction history). We adopt a multimodal large language model (MLLM) with an encoder-decoder architecture to enable vision-conditioned reasoning and text generation. In our implementation, we use the Qwen-VL series [38, 3], though the framework remains compatible with other MLLMs of similar structure. We denote the model as $\mathcal{F}_\theta$, which encodes the inputs and generates outputs via auto-regressive decoding. At time step $t$, the model receives the user instruction $\mathbf{u}$, the current screen $\mathbf{o}_t$, and the interaction history $\mathbf{h}_{t-1}$:

$$[\mathbf{c}_t, \mathbf{h}_t, \alpha_t] = \mathbf{v}_t \sim \mathcal{F}_\theta(\mathbf{u}, \mathbf{o}_t, \mathbf{h}_{t-1}) \tag{3}$$

The screen $\mathbf{o}_t$ is embedded as visual features, while $\mathbf{u}$ and $\mathbf{h}_{t-1}$ are embedded as text. These are fused in the decoder, which generates a sequence $\mathbf{v}_t \in \mathbb{D}^L$ of length $L$ auto-regressively. $\sim$ indicates that the output $\mathbf{v}_t$ is generated from the policy model $\mathcal{F}_\theta$. The output is parsed into: (1) a CoT reasoning trace $\mathbf{c}_t$, (2) an updated interaction history $\mathbf{h}_t$, and (3) the predicted GUI actions $\alpha_t$.

### 3.3 Agent Reasoning Framework

We now present our three core subtasks the agent performs at each interaction step. First, the agent engages in structured reasoning, where it processes the current input and previous history to form a coherent understanding of the task. Next, it predicts the appropriate action based on this understanding. Finally, the agent updates the history summary, integrating the new information for future decision-making. These subtasks ensure effective integration of historical context and support consistent, informed decisions throughout the interaction.

**Structured Reasoning Subtask.** To improve both decision quality and interpretability, we introduce a structured reasoning subtask that mirrors human cognitive strategies: first assessing task progress, then determining the next action. The output of this subtask $\mathbf{c}_t$, is divided into two components: *Progress Estimation* and *Decision Reasoning*. As shown in Figure 1, the agent estimates navigation progress by analyzing $\mathbf{o}_t$ and $\mathbf{h}_{t-1}$, basing its reasoning on the current observation and prior actions. Based on this, the agent determines the next action, guided by $\mathbf{u}$ and prior decisions, ensuring alignment with the task objectives. This step-wise reasoning structure promotes both coherent decision-making and interpretable intermediate reasoning.

**Action Prediction Subtask.** The action prediction subtask requires the model to predict a structured, parseable textual action based on reasoning outcomes. This textual action subsequently undergoes a parsing process to be converted into a executable data format recognizable by the agent's execution module . Specifically, the predicted textual action $\mathbf{a}_t$ is parsed into a structured, executable form $\alpha_t = \mathcal{M}(a_t) = (\alpha_t^{\text{type}}, \alpha_t^{\text{value}}, \mathcal{C})$, where: $\alpha_t^{\text{type}} \in \mathcal{V}^a$ is the action type (e.g., "CLICK", "INPUT", "ENTER"); $\alpha_t^{\text{value}} \in \mathcal{V}^v$ is the associated textual value (e.g., "Search Bar", "Date"); $\mathcal{C} = (x^{\text{pos}}, y^{\text{pos}}) \in \mathbb{R}^2$ denotes the screen coordinates of the target UI element. Here, $\mathcal{V}^a$ and $\mathcal{V}^v$ are finite sets of valid action types and values, respectively. The finiteness of these two sets ensures the controllability of the agent's action space and the validity of each generated action.

**History Summary Subtask.** To sustain long-term coherence during task execution, the agent maintains a concise yet information-dense representation of prior actions and GUI states. At each step, it summarizes $\mathbf{o}_t$, $\mathbf{h}_{t-1}$ and $\mathbf{u}$ into a concise textual memory (Figure 1). This updated hidden state $\mathbf{h}_t$ allows the agent to transcend the constraints of individual step details, continuously track task progress over extended time horizons, and thereby formulate more targeted, context-aligned decisions for future steps. In comparison to approaches relying on raw visual data [19], those utilizing feature-space-compressed historical screenshots [50] or unprocessed action-only histories [8], these semantic summaries offer two key advantages: they provide clearer hierarchical abstraction and establish tighter grounding to real-world task scenarios. Together, these strengths translate into more effective multi-step reasoning performance for the agent in complex task environments.

## 4 Training

To avoid ineffective supervision and local optima caused by sparse or low initial rewards, we design a two-stage training strategy for GUI-Rise. The first stage *cold-start training* employs supervised learning on pseudo-labeled data to establish a solid initial policy and meaningful reward signals. The second stage *reinforcement learning* refines the model with reinforcement learning, enhancing adaptability. This approach ensures stable initial training and effective fine-tuning, resulting in improved performance and generalization. An overview is shown in Figure 2.

### 4.1 Cold-start Training

In the cold-start stage, we aim to initialize the agent with essential skills in history summarization and structured reasoning. To initialize training, we use a stronger MLLM (e.g., GPT-4o-mini [52]) to generate pseudo labels, consisting of a textual summary $\mathbf{h}' \in \mathbb{D}^L$ and a structured CoT $\mathbf{c}'$.

**Pseudo-label Generation.** We employ a retrospective labeling strategy [29] to generate the agent's intermediate reasoning and history summary based on known correct actions, creating accurate and goal-consistent pseudo-labels for supervised learning. For each trajectory, pseudo-labels are generated step-by-step using GPT-4o-mini. At the initial step ($t = 0$), we input observation $\mathbf{o}_0$, user instruction $\mathbf{u}$, and ground-truth action $\alpha_0^{gt}$, producing a history summary $\mathbf{h}_0'$ and structured CoT $\mathbf{c}_0'$ as pseudo-labels. For subsequent steps ($t > 0$), we include the previous step's summary $\mathbf{h}_{t-1}'$, current observation $\mathbf{o}_t$, instruction $\mathbf{u}$, and action $\alpha_t^{gt}$, generating pseudo-labels $\mathbf{h}_t'$ and $\mathbf{c}_t'$. This sequential process accumulates history through generated summaries (labeling details and prompts are in the supplementary material B.1).

**Supervised Fine-Tuning.** During cold-start training, the ground-truth action label $\alpha^{gt}$ from trajectory annotations supervises final action prediction. These components are serialized into a target token sequence: $\mathbf{y} = \mathcal{T}([\mathbf{c}', \alpha^{gt}, \mathbf{h}']) \in \mathbb{D}^l$. Given the user instruction $\mathbf{u}$, current observation $o_t$, and previous history summary $\mathbf{h}_{t-1}$, the agent is trained to autoregressively generate this sequence using

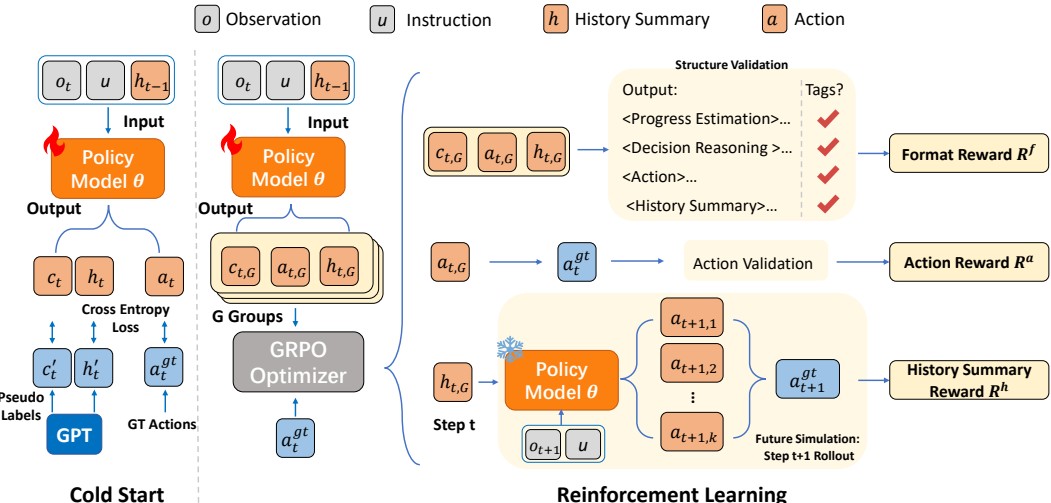

Figure 2: Overview of the GUI-Rise training pipeline. The training consists of two stages: (1) supervised learning with pseudo-labeled summaries and ground truth action trajectories to initialize reasoning, and (2) reinforcement learning with rule-based and model-based rewards to improve decision-making and generalization.

a standard token-level cross-entropy loss:

$$\mathcal{L}_{\text{CE}} = -\sum_{j=1}^{|\mathbf{y}|} \log P_\theta(\mathbf{y}_j \mid \mathbf{y}_{<j}, \mathbf{u}, \mathbf{o}_t, \mathbf{h}_{t-1}) \tag{4}$$

where $P_\theta(y_i \mid \cdot)$ denotes the probability assigned by the model's decoder over the vocabulary at position $j$.

## 4.2 Reinforcement Learning

After cold-start training, we utilize RL to further agent's policy. Specifically, we adopt the GRPO [33] algorithm, which enables reward optimization without human annotations by leveraging predefined task-specific reward functions. As illustrated in Figure 2, to guide the agent toward producing structured reasoning and correct decisions, we design three complementary reward functions: (1) a format reward function $\mathcal{R}^f$, enforcing structural correctness of outputs; (2) an action reward function $\mathcal{R}^a$, providing binary feedback on action type, format, and location; (3) a history summary reward function $\mathcal{R}^h$, measuring how well the generated summary supports accurate future actions. Together, these rewards guide the model to produce coherent reasoning traces and effective summaries for multi-step decision making.

**Format Reward.** To encourage structured output, we introduce a format reward function $\mathcal{R}^f$ based on predefined XML-style tags. To specifically promote structured CoT reasoning under this reward design, we separately assign different tags to the progress estimation and decision reasoning during the inference process. This tag-level decomposition guides the agent to generate reasoning in a modular and logically ordered manner, thereby improving interpretability and promoting step-wise decision making. The expected output format consists of four tagged components, appearing in the following fixed order: "<Progress Estimation>...</Progress Estimation>", "<Decision Reasoning>...</Decision Reasoning>", "<Action>...</Action>", and "<Memory Summary>...</Memory Summary>". We define a function, CheckTags($v_{t,i}$), that returns true if the output $\mathbf{v}_{t,i}$ strictly adheres to the prescribed tag sequence. We set the format reward $r_{t,i}^f \in \mathbb{R}$ to 1 if CheckTags($\mathbf{v}_{t,i}$) returns true, and 0 otherwise:

$$r_{t,i}^f = \mathcal{R}^f(v_{t,i}) = \begin{cases} 1 & \text{if CheckTags}(\mathbf{v}_{t,i}) == \text{true} \\ 0 & \text{else} \end{cases} \tag{5}$$

**Action Reward.** To assess the correctness and executability of predicted actions, we propose a composite action reward function $\mathcal{R}^a$. Concretely in the function, the correctness and consistency of $\alpha_{t,i}$ is then assessed based on three criteria: (i) conformity to the required structural format, (ii) consistency between the predicted action type $\alpha_{t,i}^{\text{type}}$ and the ground-truth type, and (iii) spatial accuracy, determined by whether the predicted coordinates $\mathcal{C}_{t,i}$ fall within the bounding box $b_t = (x_1^{\text{pos}}, y_1^{\text{pos}}, x_2^{\text{pos}}, y_2^{\text{pos}}) \in \mathbb{R}^4$ of the target UI element. Based on these criteria, the reward function is defined as (We add more action reward details in the supplementary materials B.2):

$$r_{t,i}^a = \mathcal{R}^a(\alpha_{t,i}, \alpha_{t,i}^{\text{gt}}, b_t) \tag{6}$$

**History Summary Reward.** The purpose of this reward is to assess the quality of the history summary by evaluating whether it contributes to correct future actions. The design avoids training a separate evaluation model by leveraging the model's own prediction behavior. Specifically, if the $i$-th action $\alpha_{t,i}$ is incorrect (i.e., $r_{t,i}^a = 0$), the history summary is considered invalid and receives no reward. Otherwise, the model performs additional $k$ rollouts using $\mathbf{h}_{t,i}$ as input to predict the next output $\hat{\mathbf{v}}_{t+1} = \mathcal{F}_\theta(\mathbf{u}, \mathbf{o}_{t+1}, \mathbf{h}_{t,i})$ with no gradient backpropagation. The history summarization reward is then computed by applying $\mathcal{R}^a$ to the predicted action $\hat{\alpha}_{t+1}$ extracted from $\hat{\mathbf{v}}_{t+1}$. This process can be defined as:

$$r_{t,i}^h = \mathcal{R}^h(\mathbf{h}_{t,i}) = \begin{cases} 0, & \text{if } r_{t,i}^a = 0 \\ \frac{1}{k}\sum_{j=1}^k \mathcal{R}^a(\hat{\alpha}_{t+1,j}, \alpha_{t+1,i}^{gt}, b_{t+1}), & \text{else} \end{cases} \tag{7}$$

This structure rewards history summaries that enable accurate future behavior while directly boosting task success rates. By linking past summaries' value to subsequent action effectiveness, it empowers the model to proactively learn and prioritize historically syntheses with tangible task relevance. Over time, this incentive drives the model to independently identify key contextual cues in historical data that improve outcomes, turning history summarization into a self-improving loop for better task success rate.

Our total reward integrates assessments of both output format correctness, action accuracy and history quality, it can be formated as :

$$r_{t,i} = r_{t,i}^f + \lambda^a \cdot r_{t,i}^a + \lambda^h \cdot r_{t,i}^h \tag{8}$$

where $\lambda^a$ and $\lambda^h$ are the weights for the action and history rewards, respectively. Based on this composite reward, we compute the advantage $A_{t,i}$ via group-level normalization as below,

$$A_{t,i} = \frac{r_{t,i} - \text{mean}(\{r_{t,i}\}_{i=1}^G)}{\text{std}(\{r_{t,i}\}_{i=1}^G)}, \tag{9}$$

and optimize the GRPO objective via policy gradient descent (More details, please refer to the supplementary materials B.3).

## 5 Experiments

We report comprehensive experimental results across various settings. Section 5.1 outlines the setup; Section 5.2 compares our method with state-of-the-art baselines in out-of-domain scenarios, while in-domain results are presented in Section 5.3. Section 5.4 evaluates zero-shot performance in an online environment. Sections 5.5 cover the ablation study and an analysis of the history representation. Further experiments—including sensitivity analysis, cold-start scenarios, impact by history representation and case studies are provided in the supplementary material C.

### 5.1 Experimental Setup

**Datasets.** Our experiments use three offline GUI navigation benchmarks and one online benchmark. (i) Mind2Web (Web) [9], comprising 2,350 unique episodes across websites and an action space with three action types. (ii) AITW (Mobile) [30], featuring an Android smartphone environment with an action space of 11 actions. (iii) GUIAct [6], a benchmark that contains both web and mobile GUI navigation tasks. (iv) MiniWob (Online) [35], an online interactive environment with two action types.

| Method | Base Model | Cross-Task | | | Cross-Website | | | Cross-Domain | | |
|---|---|---|---|---|---|---|---|---|---|---|
| | | Ele.Acc | OP.F1 | Step SR | Ele.Acc | OP.F1 | Step SR | Ele.Acc | OP.F1 | Step SR |
| *Standard Setting* | | | | | | | | | | |
| MindAct[9] | - | 55.1 | 75.7 | 52.0 | 42.0 | 65.2 | 38.9 | 42.1 | 66.5 | 39.6 |
| GPT-4[28] | - | 41.6 | 60.6 | 36.2 | 35.8 | 51.1 | 30.1 | 37.1 | 46.5 | 26.4 |
| OmniParser[24] | - | 42.4 | 87.6 | 39.4 | 41.0 | 84.8 | 36.5 | 45.5 | 85.7 | 42.0 |
| CogAgent[13] | Qwen-VL-7B[2] | 22.4 | 53.0 | 17.6 | 18.4 | 42.4 | 13.4 | 20.6 | 42.0 | 15.5 |
| Qwen-VL[2] | | 15.9 | 86.7 | 13.3 | 13.2 | 83.5 | 9.2 | 14.1 | 84.3 | 12.0 |
| SeeCilck[8] | | 28.3 | 87.0 | 25.5 | 21.4 | 80.6 | 16.4 | 23.2 | 84.8 | 20.8 |
| Qwen2-VL-2B[38] | Qwen2-VL-2B | 37.7 | 86.4 | 33.2 | 36.0 | 79.2 | 27.6 | 36.3 | 81.8 | 30.7 |
| ShowUI-2B[19] | | 39.9 | **88.6** | 37.2 | 41.6 | **83.5** | 35.1 | 39.4 | **86.8** | 35.2 |
| GUI-Rise | | **45.5** | 84.8 | **38.8** | **43.0** | 82.5 | **35.4** | **46.5** | 84.1 | **39.7** |
| Qwen2.5-VL-3B[3] | Qwen2.5-VL-3B | **52.1** | **90.2** | **48.3** | 49.8 | 85.2 | 43.5 | 48.7 | **87.3** | 44.1 |
| GUI-Rise | | 51.9 | 88.4 | 46.2 | **51.7** | **85.6** | **44.7** | **53.0** | 87.0 | **47.6** |
| *Zero-Shot Setting* | | | | | | | | | | |
| Qwen2-VL-2B[38] | Qwen2-VL-2B | 18.8 | 85.0 | 15.5 | 20.0 | 80.4 | 14.3 | 22.7 | 83.5 | 18.1 |
| ShowUI-2B[19] | | 21.4 | 85.2 | 18.6 | 21.9 | 81.9 | 16.8 | 24.4 | 83.9 | 21.4 |
| GUI-Rise | | **27.0** | **85.2** | **24.2** | **25.5** | **82.0** | **21.1** | **33.4** | **84.0** | **29.7** |
| Qwen2.5-VL-3B[3] | Qwen2.5-VL-3B | 22.2 | 87.1 | 20.1 | 22.5 | 84.3 | 17.0 | 24.8 | 84.3 | 22.8 |
| GUI-Rise | | **29.4** | **87.4** | **24.8** | **26.1** | **85.7** | **22.4** | **35.1** | **84.8** | **30.1** |

Table 1: Out-of-domain evaluation results on the Mind2Web benchmark. The table reports performance under two settings: (1) the standard setting, where models are trained on the Mind2Web training set and evaluated on its test set; and (2) the zero-shot setting, where models are trained on the GUIAct training set and evaluated on the Mind2Web test set.

It is used to complement the offline benchmarks and evaluate performance in real-time interaction. Further details for each benchmark are provided in the supplementary material.

**Settings.** To assess the generalization capability of GUI-Rise, we evaluate it under both out-of-domain and in-domain scenarios [19, 23]. For out-of-domain evaluation, we conduct generalization performance testing on both mobile and web platforms. In addition, we conduct zero-shot evaluation on the online platform MiniWob to evaluate the model's capability in handling dynamic environments. For in-domain evaluation, we train the model on the training set of AITW and test it on the respective test set.

**Evaluation Metrics.** Mind2Web is evaluated using element accuracy (Ele.Acc), operation F1 (Op.F1), and step success rate (Step SR) across three verified test splits—test-task, test-website, and test-domain—covering variations in tasks, websites, and domains. On AITW, action accuracy [30] is used to measure the per-step success rate. For MiniWob, the success rate is averaged over 50 random seeds per task and then aggregated across tasks [8].

## 5.2 Out-of-Domain Evaluation

**Mind2Web.** Given the inherently out-of-domain nature of the Mind2Web test set, we adopt two generalization evaluation settings from ShowUI [19]: standard and zero-shot. In the standard setting, the model is trained on the Mind2Web training set and evaluated on its OOD test set. In the zero-shot setting, GUI-Rise, pretrained on the GUIAct dataset, is directly evaluated on the Mind2Web OOD test set. Results are detailed in Table 1.

In the standard setting, GUI-Rise with the Qwen2-VL-2B [38] backbone achieves the highest step success rates (Step SR) across all splits, notably reaching 39.7 points in the cross-domain split. With the stronger Qwen2.5-VL-3B [3], GUI-Rise improves step SR by 1.2 and 3.5 points in the cross-website and cross-domain settings, respectively. In the zero-shot setting, GUI-Rise significantly outperforms baselines, achieving a 38.7% performance improvement over the previous state-of-the-art ShowUI in the cross-domain split. These results demonstrate GUI-Rise's superior generalization in GUI navigation tasks.

**AITW.** To evaluate generalization on the mobile platform, we evaluate GUI-Rise on the AITW benchmark using the same zero-shot setting as in Mind2Web. As shown in Table 2, GUI-Rise achieves substantial improvements over the previous state-of-the-art method ShowUI across all metrics, with a relative gain of 50.7% in the overall performance metric. Notably, for the more complex WebShop

| Method | Base Model | General | Install | G.Apps | Single | WebShop | Overall |
|--------|-----------|---------|---------|--------|--------|---------|---------|
| *In-Domain Setting* | | | | | | | |
| ChatGPT-CoT[51] | – | 5.9 | 4.4 | 10.5 | 9.4 | 8.4 | 7.7 |
| PaLM2-CoT[30] | – | – | – | – | – | – | 39.6 |
| OmniParser[24] | – | 48.3 | 57.8 | 51.6 | 77.4 | 52.9 | 57.7 |
| SeeClick [8] | Qwen-VL-7B [2] | 54.0 | 66.4 | 54.9 | 63.5 | 57.6 | 59.3 |
| Qwen2-VL-2B[38] | Qwen2-VL-2B | 61.4 | 71.8 | 62.6 | 73.7 | 66.7 | 67.2 |
| ShowUI-2B[19] | | 63.9 | 72.5 | 69.7 | 77.5 | 66.6 | 70.0 |
| GUI-Rise | | **64.4** | **73.9** | **69.7** | **78.2** | **68.2** | **71.1** |
| Qwen2.5-VL-3B[3] | Qwen2.5-VL-3B | 67.3 | 75.2 | 72.3 | 79.1 | 69.0 | 72.5 |
| GUI-Rise | | **68.4** | **76.8** | **73.1** | **80.0** | **69.9** | **73.7** |
| *Zero-Shot Setting* | | | | | | | |
| Qwen2-VL-2B[38] | Qwen2-VL-2B | 31.0 | 46.9 | 40.2 | 19.4 | 36.5 | 34.7 |
| ShowUI-2B[19] | | 32.1 | 47.7 | 42.0 | 20.1 | 37.4 | 35.9 |
| GUI-Rise | | **54.3** | **57.5** | **50.3** | **55.2** | **52.9** | **54.1** |
| Qwen2.5-VL-3B[3] | Qwen2.5-VL-3B | 35.5 | 43.1 | 41.7 | 35.0 | 39.3 | 38.9 |
| GUI-Rise | | **56.4** | **59.0** | **52.3** | **59.7** | **52.7** | **56.0** |

Table 2: Evaluation results on the AITW benchmark. The table reports performance under two settings: (1) the in-domain setting, where models are trained on the AITW training set and evaluated on its test set; and (2) the zero-shot setting, where models are trained on the GUIAct training set and evaluated on the AITW test set.

| Method | MiniWob(ZS) | MiniWob(FT) | Android World | OSWorld* |
|--------|-------------|-------------|---------------|----------|
| SeeClick [8] | 19.5 | – | – | – |
| Qwen2-VL-2B [38] | 20.8 | 66.8 | 0.0 | – |
| ShowUI-2B [19] | 27.1 | 71.5 | 7.0 | – |
| Infigui-2B [21] | – | – | 9.0 | – |
| UI-Tars-2B [25] | – | – | – | 6.5 |
| GUI-Rise-2B | **30.6** | **72.8** | **10.4** | **8.7** |

Table 3: Success rate (%) on online benchmarks. MiniWob(ZS) follows the zero-shot setting and MiniWob(FT) follows the fine-tuning setting used in ShowUI [19], and ∗ means results are evaluated on the chrome-split of OSWorld.

task, GUI-Rise achieves a +15.5 points improvement. We attribute this gain to its structured reasoning design, which enhances the model's understanding of complex environments and tasks in shopping interfaces.

## 5.3 In-Domain Evaluation

To evaluate the in-domain performance of GUI-Rise, we conduct experiments on the AITW benchmark under an in-domain setting. As shown in Table 2, using the Qwen2-VL-2B backbone, GUI-Rise achieves an overall success rate of 71.1%, outperforming all prior baselines. Compared with the ShowUI-2B, GUI-Rise achieves consistent gains across all task categories, including a 1.6-point improvement in the complex shopping scenario (68.2% vs. 66.6%). With Qwen2.5-VL-3B, GUI-Rise further exceeds baselines across all metrics, indicating that our two sub-tasks enhance execution stability and robustness in multi-step interaction tasks.

## 5.4 Online Evaluation

To validate the generalization and efficacy of our approach, we conduct comprehensive evaluations across a suite of diverse GUI navigation benchmarks. Our model, GUI-Rise, consistently demonstrates SOTA performance (Table 3). In the zero-shot scenario on the MiniWob benchmark, GUI-Rise achieves a score of 30.6, showcasing its generalizability. Our GUI-Rise-2B model reaches 72.8

| # | TST | SCoT | HS | HSR | General | Install | G.Apps | Single | WebShop. | Overall |
|---|-----|------|----|----|---------|---------|--------|--------|----------|---------|
| 1 | × | × | × | × | 61.4 | 71.8 | 62.6 | 73.7 | 66.7 | 67.2 |
| 2 | √ | × | × | × | 61.4 | 70.6 | 63.0 | 73.2 | 62.0 | 66.0 |
| 3 | × | √ | × | × | 38.0 | 48.4 | 46.9 | 36.7 | 42.9 | 42.6 |
| 4 | √ | √ | × | × | 63.2 | 73.4 | 69.5 | 76.7 | 66.0 | 69.8 |
| 5 | √ | √ | √ | × | 64.1 | 73.2 | 69.3 | 78.0 | 67.9 | 70.7 |
| 6 | √ | √ | √ | √ | **64.4** | **73.9** | **69.7** | **78.2** | **68.2** | **71.1** |

Table 4: Ablation study of GUI-Rise on the AITW benchmark. "TST" denotes the use of two-stage training with RL; "SCoT" indicates the incorporation of structured Chain-of-Thought reasoning; "HS" means using history summary as input; "HSR" refers to applying the history summary reward during RL.

in the fine-tuned scenario, confirming the effectiveness of our approach. More importantly, to test our model's capabilities in long-horizon, multi-step scenarios that mimic real-world complexity, we evaluated it on the challenging Android World and OSWorld benchmarks. As these are online environments, they feature significantly more dynamic changes, demanding a high degree of execution context consistency. Agents must be able to summarize historical interactions and accurately understand its current progress within a task. This is precisely the challenge our history summarization design is engineered to address. Our model outperforms the prior art on both the AndroidWorld and OSWorld benchmarks, demonstrating the efficacy of our design. By effectively compressing and integrating historical context, it empowers the model to make robust decisions in complex, multi-step tasks.

### 5.5 Ablation Study

To understand the contribution of key components in our model, we conduct an ablation study focusing on Two-Stage Training (TST) which measn with/without RL, Structured CoT (SCoT), History Summary (HS) and History Summary Reward (HSR). These components are critical for enhancing the model's ability to execute tasks effectively, as shown in Table 4. Using only two-stage training (SFT followed by RL) yields limited gains (Row 2), implying that small models may struggle with exploration or overfit easily. Training action with structured CoT reasoning via SFT alone (Row 3) causes a drop in success rate, indicating the challenge of learning structured reasoning using SFT. Applying RL to structured reasoning (Row 4) substantially improves performance, confirming the benefit of step-by-step reasoning. Adding historical summaries (Row 5) brings minor gains, likely due to the lack of supervision—low-quality summaries may mislead the policy and hurt learning. To address this, we introduce a history summary reward (Row 6) to reinforce semantic consistency. This yields the best overall results, showing that guiding the model toward coherent, goal-directed reasoning enhances policy learning.

## 6 Conclusion

We introduce a reasoning-enhanced framework comprising three core sub-tasks. These sub-tasks enable the GUI-Rise model with robust contextual reasoning, action coherence, and historical integration capabilities. GUI-Rise achieves state-of-the-art success rates in GUI navigation tasks, excelling in out-of-domain (OOD) scenarios. These capabilities significantly enhance stability and generalization in complex multi-step interactions, highlighting the framework's potential for building high-performance, generalizable GUI agents.

**Limitations.** A limitation of our work is that while the model is evaluated in online environments, it is trained entirely offline. This prevents the model from adapting to new scenarios or learning from its interactions in real-time. Future work could address this by exploring online learning methods, particularly by enabling the model to reflect on its successes and failures to learn directly from live interactions.

# 7 Acknowledgments

This work was supported by NSFC 62350610269, Shanghai Frontiers Science Center of Human-centered Artificial Intelligence, HPC Platform of ShanghaiTech University, MoE Key Lab of Intelligent Perception and Human-Machine Collaboration (ShanghaiTech University).

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

# A  Datasets Details

## A.1  Mind2Web

Mind2Web is a pioneering dataset designed to develop and evaluate generalist web agents capable of performing complex tasks on any website using natural language instructions. It features over 2,000 tasks across 137 websites and 31 domains, with 7,775 training actions, offering a diverse and realistic environment for training. Unlike simulated datasets, Mind2Web uses real-world websites, providing rich data like user interaction traces and webpage snapshots. A key strength lies in its three distinct test splits—cross-task, cross-website, and cross-domain—designed to rigorously evaluate generalization performance. The dataset's action space includes three core operations: CLICK, TYPE, and SELECT, capturing essential user interactions for navigating modern web complexities.

## A.2  AITW

AITW is a comprehensive Android smartphone dataset featuring 30K instructions and 715K trajectories, collected using the Android Emulator. We follow the experimental setup of ShowUI[19], dividing the data into five domains: Google Apps, Install, Web Shopping, General, and Single. The action space includes 12 actions: CLICK, TYPE, SELECT, SCROLL UP, SCROLL DOWN, SCROLL LEFT, SCROLL RIGHT, PRESS BACK, PRESS HOME, PRESS ENTER, STATUS TASK COMPLETE, and STATUS TASK IMPOSSIBLE, enabling diverse interaction analysis.

## A.3  GUIAct

GUIAct is a multi-scenario dataset designed to enhance GUI agents' knowledge, covering web and smartphone environments. It includes GUI navigation tasks split into three partitions: "web-single," "web-multi," and "smartphone," with a unified action space of 11 action types. The web dataset comprises 67K single-step and 5,696 multi-step instructions across 50 domains and 13K websites, while the smartphone dataset includes 9,157 multi-step instructions, totaling 67K training samples. Consistent with ShowUI's experimental setup, we pretrain on GUIAct for zero-shot experiments. GUIAct uses a unified action space comprising eleven key actions: CLICK, HOVER, TAP, INPUT, SCROLL, SWIPE, SELECT TEXT, COPY, ENTER, SELECT, and ANSWER, enabling agents to interact with GUI systems across web and smartphone scenarios.

## A.4  Miniwob

MiniWoB features 2000 open-ended tasks sourced from 137 real web environments. It includes dynamic GUI environments, allowing validation of a model's adaptability to dynamic settings. Each task provides high-level instructions and action trajectories, enabling agents to perform low-level keyboard and mouse actions on the Internet. The action space includes 2 actions: CLICK and TYPE.

# B  Training Details

## B.1  Pseudo-Label Generation

We utilize a retrospective labeling strategy to generate pseudo-labels for the agent's intermediate reasoning and history summary, ensuring accuracy and alignment with the intended goal using known correct actions. This process creates reliable pseudo-labels for supervised learning, generated step-by-step with GPT-4o-mini. The following are the specific prompts employed in our pseudo-label generation process, which demonstrate the structured input for each step.

```
_PROMPT_SINGLE_WEB = """"You are an AI assistant designed to simulate the model's reasoning process before
    executing a given action in a gui navigation task.  Given the task instruction, current screenshot,
    the previous history summary, the current action to be executed and thought, generate a rigorous
    chain of thought. You must strictly follow these reasoning steps:
(1) Progress Estimation: Interface Comprehension and Progress Estimation
(2) Decision Reasoning: Strategy Formulation
(3) History Summary: Update the history summary according the action you executed

### Output format:
<Progress Estimation>
... (one or two sentence)
</Progress Estimation>
<Decision Reasoning>
... (one or two sentence)
</Decision Reasoning>
<History Summary>
... (one or two sentence)
</History Summary>

###Example Input & Output
Input:
Task Instruction: Find all events taking place in New York City during the month of September.
Current Action: {{'action': CLICK, 'value': 'Apply', 'position':[0.3, 0.66]}}
Previous History Summary: The user first changed the location to New York, then set the start date to
    September 1, and set the end data to September 30.
Output:
<Progress Estimation>
The user has successfully set the location to New York and selected the date range for September 1-30, but
    the events displayed are still for March, indicating the need to apply the date filter.
</Progress Estimation>
<Decision Reasoning>
Clicking the 'Apply' button will confirm the selected date range (September 1-30) and refresh the event
    listings to show only those occurring in New York City during September.
</Decision Reasoning>
<History Summary>
The user changed the location to New York, set the date range to September 1-30, and applied the filters
    to update the event listings.
</History Summary>

###Input
Task Instruction: {_TASK}
Current Action: {_ACTION}
Thought: {_THOUGHT}
Previous History Summary: {_MEMO}
"""
```

## B.2  Action Reward Computation

To evaluate both the correctness and executability of predicted actions, we introduce a composite action reward function $\mathcal{R}^a$. This function assesses each predicted action $\alpha_{t,i}$ based on three criteria:

**Action Structural Format Reward**: Whether the output adheres to the required dictionary-style format: {"action": "ACTION_TYPE", "value": "element", "position": [x, y]} . We define a function, CheckActionF, to check the predicted action. The function first checks if the input is a dictionary. If not, it returns False. It then verifies that the dictionary contains exactly the three required keys ("action", "value", and "position") and no extra keys, the function return true:

$$r_{t,i}^{\mathrm{af}} = \mathcal{R}^{\mathrm{af}}(\alpha_{t,i}) = \begin{cases} 1 & \text{if CheckActionF}(\alpha_{t,i}) == \text{true} \\ 0 & \text{else} \end{cases} \tag{10}$$

**Action Type Reward**: We evaluate whether the predicted action type $\alpha_{t,i}^{\mathrm{type}}$ matches the ground-truth type. Specifically, we check if the predicted value exactly equals the annotated action type for the

current step. If they match, the prediction is considered correct:

$$r_{t,i}^{\text{type}} = \mathcal{R}^{\text{type}}(\alpha_{t,i}^{\text{type}}, \alpha_{t,i}^{\text{gt type}}) = \begin{cases} 1 & \alpha_{t,i}^{\text{type}} = \alpha_{t,i}^{\text{gt type}} \\ 0 & \text{else} \end{cases} \tag{11}$$

**Action Position Reward**: We evaluate whether the predicted coordinates $\mathcal{C}_{t,i}$ fall within the bounding box $b_t = (x_1^{\text{pos}}, y_1^{\text{pos}}, x_2^{\text{pos}}, y_2^{\text{pos}}) \in \mathbb{R}^4$ of the target UI element. Specifically, we check if the predicted point lies inside the rectangular region defined by $b_t$:

$$r_{t,i}^{\text{pos}} = \mathcal{R}^{\text{pos}} = \begin{cases} 1 & \text{if } \mathcal{C} \text{ in } b_t \\ 0 & \text{else} \end{cases} \tag{12}$$

The final action reward $r_{t,i}^a$ is computed as a weighted sum of the three components:

$$r_{t,i}^a = r_{t,i}^{\text{af}} + \lambda^{\text{type}} \cdot r_{t,i}^{\text{type}} + \lambda^{\text{pos}} \cdot r_{t,i}^{\text{pos}} \tag{13}$$

### B.3 Reinforcement Learning Objective

In the second stage of our two-phase training, we adopt GRPO [33]. GRPO improves upon traditional Proximal Policy Optimization (PPO) [31, 12, 34, 45], surpasses traditional PPO by eliminating the need for a separate critic model. At time step $t$, the advantage $\hat{A}_{i,t}$ corresponding to the $i$-th output $v_i$ can be derived from Eq. (9). Then, the overall objective is:

$$b_{t,i,j}(\theta) = \frac{\pi_\theta(v_{t,i,j} \mid \mathbf{u}, \mathbf{o}_t, \mathbf{h}_{t-1}, v_{t,i,<j})}{\pi_{\theta_{\text{old}}}(v_{t,i,j} \mid \mathbf{u}, \mathbf{o}_t, \mathbf{h}_{t-1}, v_{t,i,<j})}. \tag{14}$$

$$\mathcal{J}_{\text{GRPO}}(\theta) = \mathbb{E}_{(\mathbf{u}, \mathbf{o}_t, \mathbf{h}_{t-1}, \alpha_t) \sim \mathcal{D}, \{v_{t,i}\}_{i=1}^G \sim \pi_{\theta_{\text{old}}}(\cdot \mid \mathbf{u}, \mathbf{o}_t, \mathbf{h}_{t-1})}$$
$$\left[ \frac{1}{G} \sum_{i=1}^G \frac{1}{|v_{t,i}|} \sum_{j=1}^{|v_{t,i}|} \left( \min\left( b_{t,i,j}(\theta)\hat{A}_{i,t}, \text{clip}(b_{t,i,j}(\theta), 1-\epsilon, 1+\epsilon)\hat{A}_{i,t} \right) - \beta D_{\text{KL}}(\pi_\theta \parallel \pi_{\text{ref}}) \right) \right] \tag{15}$$

where $(\mathbf{u}, \mathbf{o}_t, \mathbf{h}_{t-1}, \alpha_t)$ is a question-answer pair from the data distribution $\mathcal{D}$, $\epsilon$ is the clipping range of importance sampling ratio. To ensure stable policy updates, GRPO also introduces a KL-divergence regularization term that penalizes deviation from the reference policy distribution, and $\beta$ is a coefficient controlling the strength of the regularization. This formulation helps constrain policy updates, stabilizing training and encouraging consistency with previously learned behaviors.

## C  Experimental Analysis

### C.1  Scalability Evaluation on Larger-Scale VL Models

Previous GUI-Rise evaluations only used small VL models (2B–3B params), limiting real-world relevance—larger 7B-scale models are more practical for complex GUI tasks due to stronger reasoning. Thus, we tested Qwen2.5-VL-7B (GUI-optimized 7B VL model) on AITW and Mind2Web under SFT. Tables 5 and 6 show GUI-Rise-7B outperforms the baselines. On AITW, GUI-Rise-7B boosts overall Step SR by 3.0% with the biggest gain in Install, proving cross-task GUI effectiveness. On Mind2Web, it achieves strong cross-scenario gains Step SR. 7B experiments confirm GUI-Rise's scalability and boosts its practical value for GUI agents.

### C.2  Impact of Cold Start

In this section, we analyze the impact of the first stage in our two-stage training framework: cold start pretraining. We compare two setups on the Mind2Web and AITW datasets: (1) Cold Start + RL and (2) RL Only. The initial model is Qwen2-VL-2B. We exclude the history summary reward in this experiment, as it depends on future actions and is unrelated to cold start effects.

| Method | Base Model | Cross-Task | | | Cross-Website | | | Cross-Domain | | |
|---|---|---|---|---|---|---|---|---|---|---|
| | | Ele.Acc | OP.F1 | Step SR | Ele.Acc | OP.F1 | Step SR | Ele.Acc | OP.F1 | Step SR |
| *In-Domain Setting* | | | | | | | | | | |
| Qwen2.5-VL-7B | Qwen2.5- | 54.5 | 90.5 | 50.5 | 53.5 | 88.7 | 46.9 | 51.1 | 89.1 | 46.8 |
| Gui-Rise-7B | VL-7B | **56.9** | **90.4** | **52.3** | **56.7** | **88.7** | **50.7** | **56.4** | **90.2** | **51.4** |

Table 5: Evaluation results on the Mind2Web benchmark with 7B models.

| Method | Base Model | General | Install | G.Apps | Single | WebShop | Overall |
|---|---|---|---|---|---|---|---|
| *In-Domain Setting* | | | | | | | |
| Qwen2.5-VL-7B | Qwen2.5- | 68.6 | 77.4 | 76.2 | 80.0 | 68.5 | 73.7 |
| Gui-Rise-7B | VL-7B | **70.1** | **80.2** | **78.6** | **81.4** | **69.9** | **75.7** |

Table 6: Evaluation results on the AITW benchmark with 7B models.

Figure 3 and 4 shows the reward curves during training, revealing distinct patterns across datasets. On AITW, the action reward gap is minor at first and narrows over time, stabilizing at a difference of just 1.5 points. In contrast, on Mind2Web, cold start yields a sharp initial gain in action-type reward, while the action-position reward remains negligible without it. In fact, for the RL-only model, the position reward stays near zero throughout training, resulting in vanishing advantage estimates and ineffective gradient updates. Format reward improves rapidly on both datasets; within 30 iterations, the model consistently outputs responses in the correct format.

We attribute the differing outcomes to task complexity. Mind2Web's visually rich web interfaces (e.g., higher resolution, dense layouts) present a steeper learning curve than AITW's simpler mobile environments. Without a cold start, the model fails to receive meaningful reward signals early on, leading to ineffective learning.

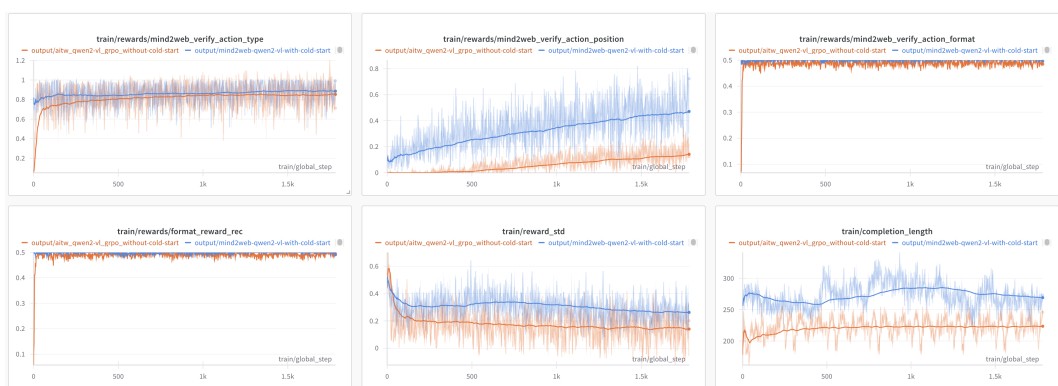

Figure 3: GUI-Rise training process on Mind2Web benchmark.

## C.3  Impact of History Representation

Effective history representation is essential for efficient and accurate GUI navigation. We evaluate five strategies—Action-Only [8], Action+Screenshot [19], ShowUI [19], GUI-Odyssey [23], and our proposed history summary representation—to assess their trade-offs between navigation success and input token efficiency. Our experimental results shown in the Figure 5, our method achieves the highest navigation success rate with the lowest token footprint. Compared to vision-heavy approaches like ShowUI, it reduces visual token overhead while improving performance. Relative to GUI-Odyssey, it further shortens input length and enhances success rates. These results highlight the importance of optimized history representation for enabling practical, high-performance GUI interaction in MLLMs.

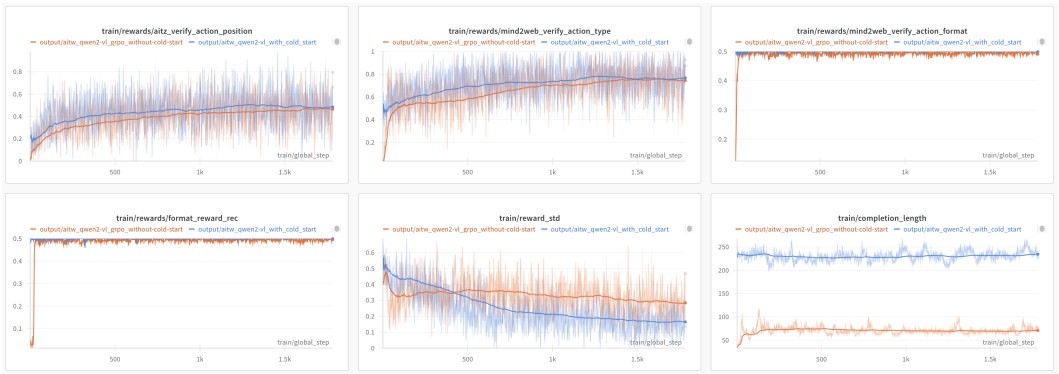

Figure 4: GUI-Rise training process on AITW benchmark.

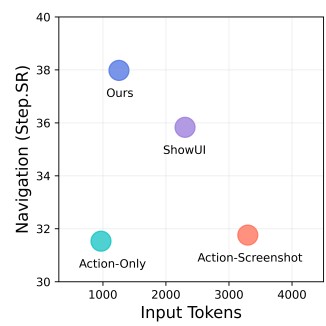

(a) Zero-shot navigation comparison between different history methods in terms of input token cost.

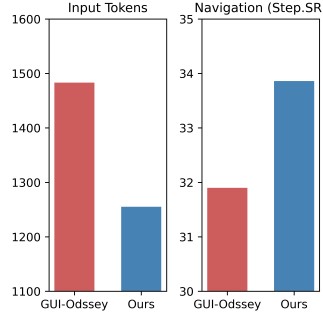

(b) Zero-shot navigation comparison between GUI-Rise and GUI-Odyssey in terms of input token cost.

Figure 5: Impact by different history representation in GUI navigation in Mind2Web benchmark.

## C.4 Case Study

We selecte several samples from the test results of two distinct platforms to conduct a case study. As illustrated in Figure 6, the agent demonstrates the capability to perform structured reasoning, analyzing the progress of the current task and reasoning about decisions through a step-by-step analytical process. Additionally, Figure 7 shows how the agent summarizes historical information across the entire trajectory, which enables coherent reasoning in future steps.

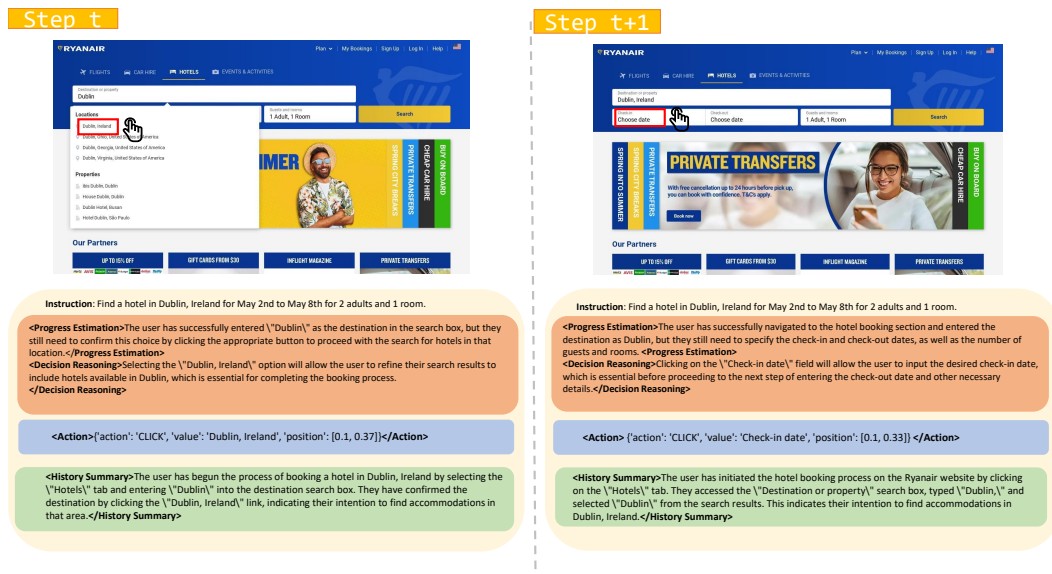

Figure 6: A case study from the Mind2Web dataset illustrates detailed reasoning across two consecutive steps.

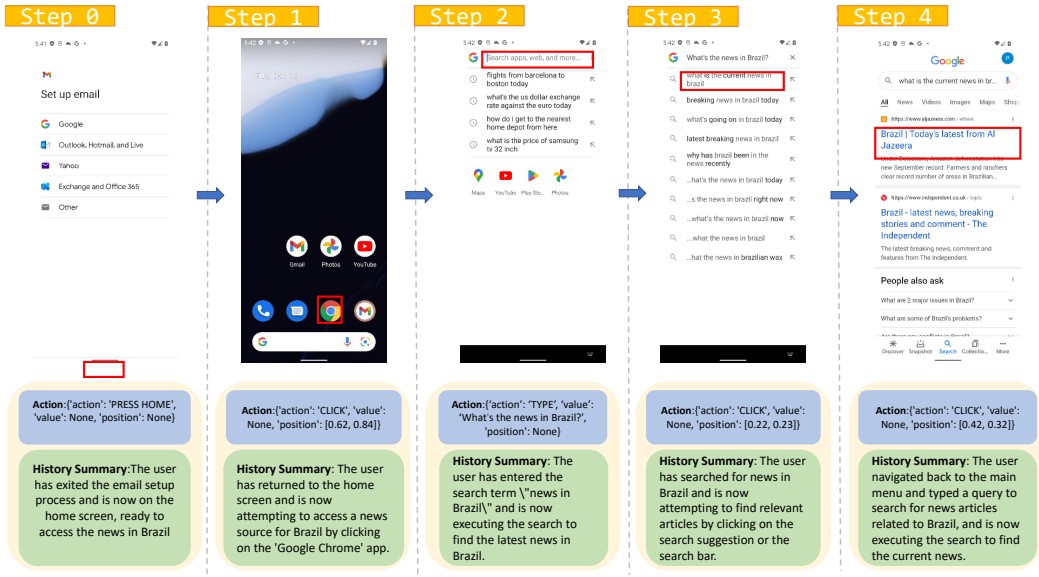

Figure 7: A case study from the AITW dataset illustrates the detailed history summarization for an entire trajectory

| | TST | SCoT | HS | HSR | General | Install | G.Apps | Single | WebShop. | Overall | Gain |
|---|---|---|---|---|---|---|---|---|---|---|---|
| 1 | × | × | × | × | 61.4 | 71.8 | 62.6 | 73.7 | 66.7 | 67.2 | - |
| 2 | √ | × | × | × | 61.4 | 70.6 | 63.0 | 73.2 | 62.0 | 66.0 | −1.2 |
| 3 | × | √ | × | × | 38.0 | 48.4 | 46.9 | 36.7 | 42.9 | 42.6 | −24.6 |
| 4 | √ | √ | × | × | 63.2 | 73.4 | 69.5 | 76.7 | 66.0 | 69.8 | +2.6 |
| 5 | √ | √ | √ | × | 64.1 | 73.2 | 69.3 | 78.0 | 67.9 | 70.7 | +0.9 |
| 6 | √ | √ | √ | √ | **64.4** | **73.9** | **69.7** | **78.2** | **68.2** | **71.1** | +1.3 |

Table 7: Ablation study of GUI-Rise on the AITW benchmark.

