# OpenReview forum: "GUI-Rise: Structured Reasoning and History Summarization for GUI Navigation"
_NeurIPS.cc/2025/Conference — NeurIPS 2025 poster_

### Official Review · Reviewer_hw3E · 2025-06-23

**Clarity:** 3
**Significance:** 2
**Originality:** 3
**Rating:** 4
**Confidence:** 5

**Summary:**

This paper proposes a strategy for training multimodal LLMs on GUI agent tasks. The proposed strategy consists of two steps. In the first step, the authors augment existing GUI trajectories with reasoning traces generated from GPT-4o-mini, so that the model can learn a structured reasoning process. In the second step, the authors uses reinforcement learning (GRPO) to continue train the model. The reward is based on step-level action correctness, and is designed to enhanced the reasoning structure. The authors use Qwen VL models as the base model, and show improvement on several mobile/web GUI agent benchmarks.

**Questions:**

In order for me to raise my score, I would like to see if each stage of the proposed method (SFT+RL) can achieve improvement on a more realistic online evaluation benchmark.

**Ethical Concerns:**

["NO or VERY MINOR ethics concerns only"]

**Final Justification:**

Thanks the authors for the response and the online evaluation. I raised my score to borderline accept.

**Limitations:**

yes

**Quality:**

2

**Strengths And Weaknesses:**

Strengths:
- The paper is well-structured with clear motivation. The method is simple and clearly explained.
- The reward design fits the proposed structured reasoning strategy.

Weakness:
- The proposed training method requires ground-truth trajectories. This makes it difficult to scale and generalize to more complex GUI tasks.
- The proposed RL is done on a static dataset, and the reward is mostly based on step-level action correctness. Thus the RL stage may not add much value after SFT. Ideally, RL should be done in an online environment, where the agent can learn by interacting with the environment.
- The evaluation benchmarks are mostly offline, and the only online evaluation is done on a small dataset with few baselines. In order to show that the proposed method can work on more realistic environments, the authors need to evaluate on more challenging online benchmarks, such as WebArena, OSWorld, etc.
- The authors have mentioned online training and evaluation as future work. However, I believe that it is critical that a GUI agent can work in a dynamic and realistic environment, rather than just static datasets.

---

> ### Author Rebuttal · Authors · 2025-07-31
>
> We appreciate the reviewer's feedback and address the main concerns below.
>
> ---
>
> ### W1:
>
> Our current training approach, which leverages ground-truth trajectories, can be viewed as a form of imitation learning. We frame our current work as a first stage toward addressing complex GUI tasks.  The primary goal of this stage is to establish a strong foundation by enabling the agent to acquire basic operational skills from a high-quality dataset. Without the guidance of ground-truth trajectories, learning from sparse rewards alone (especially in the early stages) would lead to inefficient reinforcement learning and significantly slower convergence.
>
> We argue that exploring this foundational stage is itself an important and standalone research contribution. It provides a reliable starting point for building agents capable of tackling more complex and realistic GUI tasks.
>
> To address scalability in the later stages, once the model has acquired essential skills through imitation learning, we further refine it using reinforcement learning within a dedicated simulator. In this environment, the agent no longer relies on step-by-step demonstrations; instead, it receives reward signals based solely on the final task outcome.
>
> ---
>
> ### W2:
>
> We respectfully disagree with the assertion that "the RL stage may not add much value." As shown in Table 1 on the Mind2Web benchmark and Table 2 on the AITW benchmark in the manuscript, our method achieves significant improvements in zero-shot task success rate by 31.5% and 50.7%, respectively. These substantial gains clearly demonstrate the added value of our RL stage beyond supervised fine-tuning.
>
> That said, we agree that performing RL in an online interactive environment is a promising direction for future work, and we consider it complementary to our current setting.
>
> ---
>
> ### W3:
>
> Thanks for your suggestions, we have added more realistic environmental results in the table below.
> For WebArena, it utilizes Axtree and SoM annotations to guide action execution, requiring agents to provide element IDs rather than click coordinates. As our vision-based agent produces click coordinates, it is fundamentally incompatible with WebArena’s evaluation framework. Instead, we evaluate our agent on the Android World online benchmark, which comprises 116 tasks across 20 real-world mobile applications. Our agent demonstrates a 16% performance improvement over the prior 2B-parameter state-of-the-art model, InfiGUIAgent. Detailed results are presented in the table below, further confirming its efficacy in complex GUI navigation scenarios.
>
> Additionally, we evaluate our agent on the OS-World benchmark (chrome-split), with findings reported in the supplementary material Table 3. Besides, to align with prior works such as ShowUI, we supplement our MiniWoB experiments under the supervised fine-tuning setting.
>
> Collectively, these outcomes highlight that our approach sustains top-tier performance and robust generalization across diverse, challenging online interactive environments.
>
> **Table 4-1. Step success rate (%) of 2B-parameter models on online benchmarks.**
>
> | Method        | MiniWob | Android World | OS World |
> |--------------|---------|----------------|----------|
> | Qwen2-VL-2B  | 66.8    | 0.0            | --       |
> | ShowUI-2B    | 71.5    | 7.0            | --       |
> | Infigui-2B   | --      | 9.0            | --       |
> | UI-Tars-2B   | --      | --             | 6.5      |
> | GUI-Rise-2B  | **72.8**    | **10.4**           | **8.7**      |
>
> ---
>
> ### W4:
>
> We clarify that in the limitations section of the paper, we specifically referred to online training as a direction for future exploration, not online evaluation.
>
> That said, we fully agree with the reviewer on the importance of evaluating GUI agents in dynamic and realistic environments. To this end, we have already conducted extensive online evaluations across diverse environments, including MiniWoB (Table 4 in the manuscript), OS-World (Table 4 in the supplementary), and Android World (Table 4-1), demonstrating the model’s effectiveness in dynamic and realistic settings.
>
> To avoid such misunderstandings, we will revise the limitations section in the final version to more clearly distinguish between online evaluation and online training.
>
> ---

---

> > ### Comment · Area_Chair_ctNx · 2025-08-04
> > **Please respond to the author's rebuttal post**
> >
> > Hi Reviewer hw3E, I see no response letting me know whether or not the rebuttal has changed your opinion. Could you please let me and the authors know by engaging? This process is critical to enabling the (S)ACs to make a decision on this work.
> >
> > --Your AC

---

### Official Review · Reviewer_Dt5U · 2025-06-28

**Clarity:** 3
**Significance:** 3
**Originality:** 3
**Rating:** 4
**Confidence:** 4

**Summary:**

This paper investigates the application of Large Language Models in GUI Navigation and proposes the GUI-Rise method. The core components include structured reasoning, action prediction, and history summarization. Moreover, GUI-Rise employs the GRPO method for policy optimization. The authors conduct experiments on three benchmarks to validate the effectiveness of their approach.

**Questions:**

See Weaknesses

**Ethical Concerns:**

["NO or VERY MINOR ethics concerns only"]

**Final Justification:**

The authors have addressed all my concerns, and I maintain my positive rating.

**Limitations:**

Yes

**Quality:**

3

**Strengths And Weaknesses:**

**Strengths**:

1. The paper is well-organized and clearly written, with figures and tables effectively illustrating the core concepts.

2. The authors provide comprehensive experimental validation including Out-of-Domain, In-Domain, Online, and Ablation studies to demonstrate GUI-Rise's effectiveness.

3. The authors include case studies in the appendix to enhance understanding of GUI-Rise.

**Weaknesses**:

1. The paper primarily uses GRPO for model optimization but does not sufficiently justify its necessity in the UI context. Specifically, the authors should discuss whether GUI-Rise could utilize alternative methods like PPO or DAPO for policy optimization.

2. GUI-Rise is currently validated only on 2B and 3B models. Testing on larger models, such as Qwen-VL-7B, would strengthen the evaluation.


3. Typos: in Figure 1, \<Decision Reasoning\> -> </Decision Reasoning>

4. Typos: Notation inconsistency between equations (7) and (6), e.g., in (7), $\mathcal{R}^a$ -> $R^a$

---

> ### Author Rebuttal · Authors · 2025-07-31
>
> We thank reviewer for the constructive comments. We provide our feedbacks as follows.
>
> ---
>
> ### W1:
>
> The reason we adopt GRPO lies in its training efficiency, as it avoids the computational overhead associated with critic models. In contrast, actor-critic methods like PPO require optimization of both actor and critic models, leading to significantly higher resource consumption.
>
> Regarding DPO, our preliminary experiments revealed its sensitivity to preference pair quality. For instance, using Qwen2-VL-2B, we constructed around 2,000 preference pairs on the Mind2Web dataset. The model trained with these preferences achieved step SR of only 28.5%, 25.1%, and 27.6% on the cross-task, cross-website, and cross-domain splits, respectively, while achieving 68.4% on the training preference pairs itself. This indicates a tendency for DPO to overfit in GUI-based tasks, thus compromising generalization performance.
>
> We appreciate the reviewer’s suggestion to consider alternatives such as DAPO or other methods (as RLOO and REMAX). These methods share a similar principle with GRPO: they aim to obtain reliable advantage estimates without relying on a critic model. In particular, DAPO can be seen as an improvement over GRPO. Adopting these methods is indeed feasible. However, we would like to emphasize that the choice of training algorithm is orthogonal to our main contribution. Our work primarily focuses on proposing a framework that integrates structured reasoning, action prediction, and historical summarization to enhance GUI agent performances. An analysis of such optimization strategies would be a valuable direction for future work.
>
> ---
>
> ### W2:
>
> Following the reviewers’ suggestion, we additionally trained the larger Qwen2.5-VL-7B model on the training sets of AITW and Mind2Web, and evaluated it on their respective test sets. We ensured a fair comparison with Qwen2.5-VL-7B baseline by using the same training sets, as shown in Tables 3-1 and 3-2. The step success rate results, demonstrate consistent improvements over the baseline by 3% on AITW and 10% on Mind2Web. It further confirming both the effectiveness and scalability of our method.
>
> **Table 3-1. Step success rate (%) of Qwen2.5-VL-7B (SFT) and GUI-Rise-7B on the AITW benchmark.**
>
> | Model                | General | Install | G.Apps | Single | WebShop | Overall |
> |---------------------|---------|---------|--------|--------|---------|---------|
> | Qwen2.5-VL-7B (SFT) | 68.6    | 77.4    | 76.2   | 80.0   | 68.5    | 73.7    |
> | GUI-Rise-7B         | **70.1**    | **80.2**    | **78.6**   | **81.4**   | **69.9**    | **75.7**    |
>
> ---
>
> **Table 3-2. Performance (%) of Qwen2.5-VL-7B (SFT) and GUI-Rise-7B on the Mind2Web benchmark.**
>
> | Model                | Cross-task         |                     |           &nbsp;&nbsp;           | Cross-website       |                     |             &nbsp;&nbsp;         | Cross-domain        |                     |           &nbsp;&nbsp;           |
> |----------------------|--------------------|---------------------|----------------------|----------------------|---------------------|----------------------|----------------------|---------------------|----------------------|
> |                      | Ele.Acc            | OP.F1       &nbsp;&nbsp;&nbsp;&nbsp;        | Step SR        &nbsp;&nbsp; &nbsp;&nbsp;    | Ele.Acc             | OP.F1      &nbsp;&nbsp;&nbsp;&nbsp;         | Step SR      &nbsp;&nbsp; &nbsp;&nbsp;       | Ele.Acc             | OP.F1       &nbsp;&nbsp;&nbsp;&nbsp;        | Step SR      &nbsp;&nbsp; &nbsp;&nbsp;       |
> | Qwen2.5-VL-7B (SFT)  | 54.5               | 90.5                | 50.5                | 53.5                | 88.7                | 46.9                 | 51.1                | 89.1                | 46.8                 |
> | GUI-Rise-7B          | **56.9**               | **90.4**                | **52.3**                | **56.7**                | 88.7                | **50.7**                 | **56.4**                | **90.2**                | **51.4**                 |
>
> ---
>
> ### W3 & W4:
>
> We are grateful to the reviewer for their careful attention. These typos will be fully corrected in the final manuscript.
>
> ---

---

> > ### Comment · Reviewer_Dt5U · 2025-08-05
> >
> > Thanks for the detailed responses. I will maintain my positive score.

---

> > > ### Author Response · Authors · 2025-08-05
> > >
> > > We sincerely appreciate your recognition of our efforts and are grateful for your valuable suggestions, which we will incorporate into our revisions.
> > > Thank you!

---

### Official Review · Reviewer_W4us · 2025-07-01

**Clarity:** 2
**Significance:** 2
**Originality:** 2
**Rating:** 4
**Confidence:** 5

**Summary:**

This paper proposed a reasoning-enhanced framework that integrates structured reasoning, action prediction, and history summary.
The proposed GUI agent, GUI-Rise, is trained by a two-stage training scheme, including SFT and RL. The propose framework structures the reasoning and execution pipeline for GUI-based tasks, introducing a history representation mechanism to improve decision accuracy. The experimental results show that the proposed GUI-Rise achieves SOTA results on standard benchmarks.

**Questions:**

The current discussion regarding methods for encoding history lacks sufficient rigor. For instance, the assertion in Lines 34-36 states that "Existing systems encode history either (i) as action sequences alone [8, 9], which omit visual state and hinder progress estimation, or (ii) as full-resolution screenshots [19, 22, 28], which are computationally expensive and force severely truncated context windows." This claim overlooks alternative approaches, such as those adopted by UI-Hawk, which utilize low-resolution history screenshots instead of full-resolution ones. Consequently, the presented motivation for new encoding methods may not be as persuasive as intended, as it doesn't fully account for the spectrum of existing solutions.

**Ethical Concerns:**

["NO or VERY MINOR ethics concerns only"]

**Final Justification:**

The rebuttal addressed some of my concerns and I would like to change the rating to borderline accept after reading the rebuttal.

**Limitations:**

Yes

**Paper Formatting Concerns:**

--

**Quality:**

2

**Strengths And Weaknesses:**

## Strengths
This paper proposed a reasoning-enhanced framework. The experimental results show that the proposed framework is effective.

## Weaknesses
1. The core contributions of the proposed three subtasks primarily reside in prompt engineering. Consequently, the technical novelty and architectural innovation appear somewhat limited.
2. The evaluation of the proposed history summary subtask is constrained to comparisons with raw visual or action-only histories. A more comprehensive analysis should include state-of-the-art approaches such as UI-Hawk [1], which utilizes low-resolution visual history images. This alternative method also achieves a commendable balance between computational efficiency and task accuracy, and its exclusion weakens the comparative rigor of the presented results.

[1] UI-Hawk: Unleashing the screen stream understanding for gui agents

---

> ### Author Rebuttal · Authors · 2025-07-31
>
> We appreciate the reviewer's feedback and address the main concerns below.
>
> ---
>
> ### W1:
>
> We respectfully disagree with the claim that our core contributions mainly reside in prompt engineering. In fact, the effectiveness of prompt engineering depends on the model’s inherent capabilities. However, our experiments on Mind2Web demonstrate that smaller models (e.g., 2B/3B parameters) often fail to produce meaningful history summaries or accurate progress estimations. When directly prompting the model to output such information, we frequently observe vague or noisy history summaries, such as: *The user started planning a trip; The starting point was set to Braintree with a typo in the "From" field; Although they interacted with some additional elements, their effect on the progress is not explicitly clear.*
> These noisy outputs negatively impact downstream reasoning, and in our experiments, such direct prompting caused a 20% drop in step success rate on the Mind2Web benchmark.
>
> Our key contribution to our method is not in designing handcrafted prompts, but in developing a structured reasoning framework that enables the model to autonomously learn to perform each subtask during training. We view this dynamic subtask learning framework as a core technical innovation that enhances the model’s reasoning capabilities, distinguishing it from conventional prompt engineering approaches.
>
> This design leads to consistent performance gains across both offline and online benchmarks (as shown in Tables 1, 2, and 4 in the manuscript), achieving new state-of-the-art results. These results validate both the practical effectiveness and the technical significance of our method.
>
> ---
>
> ### W2 & Q1:
>
> We appreciate the reviewers' comments and suggestions for improving the rigor of our statements and address their main concerns as follows.
>
> UI-Hawk reduces visual history images to one-quarter of their original size and adopts a visual token compression ratio of 16, enhancing computational efficiency and performance. However, low-resolution images may lose detailed information, such as small interactive button states, compromising historical data integrity. Furthermore, a key limitation of UI-Hawk is that it has only been evaluated on GUI-Odyssey and its self-proposed FunUI benchmarks. It has not been tested on mainstream benchmarks widely adopted in the GUI navigation field, such as AITW and Mind2Web. Moreover, a direct and fair comparison with UI-Hawk is not feasible, as its results are reported on a privately re-annotated version of GUI-Odyssey (GUI-Odyssey+), with no public code, checkpoints, or benchmark details available.
>
> Regarding Question 1, our statement was intended to address the broader category of methods that rely on history screenshots. Specifically, approaches like UI-Hawk that process these visual sequences inherently face a trade-off between preserving information integrity and computational efficiency. Our agent, GUI-Rise, leverages the history summary capability acquired through training to self-summarize historical task states and executed actions. This approach ensures the integrity of historical information while also reducing computational overhead.
>
> We will expand the discussion of these trade-offs and related methods in the revised Related Work section.
>
> ---

---

> > ### Author Response · Authors · 2025-08-05
> >
> > We sincerely appreciate your time and thoughtful review of our manuscript. Following our rebuttal, we kindly seek confirmation on whether the revisions adequately addressed your concerns, or if we should provide any additional clarifications before the discussion deadline. Your feedback means a lot to us.

---

> > ### Comment · Reviewer_W4us · 2025-08-05
> >
> > Thanks for the authors' clarification. I would like to change the rating to borderline accept after reading the rebuttal.

---

> > > ### Author Response · Authors · 2025-08-05
> > >
> > > We sincerely appreciate the reviewer's recognition of our work and the increased score.
> > >
> > > Thank you!

---

### Official Review · Reviewer_1riz · 2025-07-03

**Clarity:** 3
**Significance:** 2
**Originality:** 3
**Rating:** 4
**Confidence:** 3

**Summary:**

This work presents a framework to train GUI agents through Cold Start SFT + Reinforcement Learning with GRPO. They predefine a structured reasoning template with 3 parts: structured reasoning, action prediction, and history summarization, which helps the agent to do better reasoning and issue correct actions. They evaluate the trained model on four different GUI navigation benchmarks and show that their method improves the base model a lot.

**Questions:**

1. Why do you design a separate history summary reward, but no separate rewards for progress estimation/decision reasoning
2. Why do you choose to put the history summary after action prediction and prompt it for the next step rather than before the action prediction to be a part of structured reasoning?

**Ethical Concerns:**

["NO or VERY MINOR ethics concerns only"]

**Final Justification:**

The authors explain in detail in response to my reviews and solve most problems. So I keep my rating of "borderline accept".

**Limitations:**

Yes

**Paper Formatting Concerns:**

No such concerns

**Quality:**

3

**Strengths And Weaknesses:**

Strengths:

1. The framework design (SFT + GRPO) is intuitive and has been proven effective on coding and math problems. This is a nice trial on GUI agent training.
2. GUI-Rise improves the base agent on GUI navigation tasks

Weaknesses:

1. The online navigation evaluation part only uses MiniWob as the benchmark, which is a relatively easy environment. There are some harder online interactive environments like WebArena [1], OS-World [2] that are commonly used for GUI agents. Lack of evaluation on these complex online environments and comparison with other baselines on such environments would make the method less persuasive
2. The improvement under the standard setting is minor on Mind2Web and AITW, which seems to indicate that, compared to vanilla SFT on the training set, your framework does not show much advantage. But you can argue and explain more about this result.
3. I have a few questions about the framework design. See the Questions part

[1] WebArena: A Realistic Web Environment for Building Autonomous Agents

[2] OSWorld: Benchmarking Multimodal Agents for Open-Ended Tasks in Real Computer Environments

---

> ### Author Rebuttal · Authors · 2025-07-31
>
> We thank reviewer for the constructive comments. We provide our feedbacks as follows.
>
> ---
>
> ### W1:
> We appreciate your suggestions and have now included more realistic environmental results in Table 1-1.
> For WebArena, it employs Axtree and SoM annotations for action execution, requiring agents to output element IDs instead of click coordinates. As our agent is a vision-based model that generates click coordinates, it is inherently incompatible with the evaluation framework of WebArena.  Instead, we conduct an evaluation on the Android World online benchmark, which consists of 116 tasks across 20 real-world mobile apps. Our agent achieves a 16% performance improvement over the previous 2B-parameter SOTA model, InfiGUIAgent, further validating its effectiveness in complex GUI navigation scenarios, with results presented in the table below.
>
> Additionally, we evaluate our agent on the OS-World benchmark (chrome-split), with results reported in the supplementary material Table 3. Moreover, to align with prior works (e.g., ShowUI), we supplement our MiniWoB experiments under the supervised fine-tuning setting. Moreover, to align with prior works such as ShowUI, we supplement our MiniWoB experiments under the supervised fine-tuning setting.
>
> Overall, these results collectively demonstrate that our approach maintains state-of-the-art performance and strong generalization across diverse and challenging online interactive environments.
>
> **Table 1-1. Step success rate (%) of 2B-parameter models on online benchmarks.**
>
> | Method        | MiniWob | Android World | OS World |
> |--------------|---------|----------------|----------|
> | Qwen2-VL-2B  | 66.8    | 0.0            | --       |
> | ShowUI-2B    | 71.5    | 7.0            | --       |
> | Infigui-2B   | --      | 9.0            | --       |
> | UI-Tars-2B   | --      | --             | 6.5      |
> | GUI-Rise-2B  | **72.8**    | **10.4**           | **8.7**      |
>
> ---
>
> ### W2:
>
> We agree with the observation that GUI-Rise yields a minor improvement in performance over a vanilla SFT baseline under the in-domain settings of Mind2Web and AITW. Similar trends [1] have been observed in other vision-language domains, such as Vision-Language Navigation, where the performance difference between RL-based methods and SFT remains small in in-domain scenarios. This is largely attributable to the strong fitting capability of SFT to the training data distribution.
>
> The core of our work lies in improving the model's generalization ability in more challenging out-of-distribution (OOD) scenarios that are closer to real-world applications. When the evaluation scenario shifts from standard settings to OOD settings, the performance of vanilla SFT is substantially degraded. In contrast, the unique design of our framework helps the model better understand and adapt to unknown tasks, resulting in performance improvements of 31.5% (Table 1 in the manuscript) and 50.7% (Table 2 in the manuscript). This demonstrates the critical value of our work for building GUI agents that can operate reliably in diverse and dynamic environments.
>
> [1] SFT Memorizes, RL Generalizes: A Comparative Study of Foundation Model Post-training
>
> ---
>
> ### Q1:
>
> Our reward design represents a trade-off: we introduce a dedicated reward specifically for the history summary subtask, while only two types of rewards, format reward and action reward, are designed for the other components. This choice balances model performance and training efficiency. Our framework can be readily extended to incorporate separate rewards for progress estimation and decision reasoning. However, we found that doing so significantly increases the computational complexity during training, while yielding only marginal performance gains.
>
> To verify its specific impact in practice, we conducted a small-scale comparative experiment on the AITW benchmark. The experimental results show that if independent rewards are designed for all components, the overall training duration of the model will nearly double, while the resulting improvement in final performance is marginal (about 1.7% improvement). As a trade-off, we chose to retain a separate reward only for the history summary component to balance performance gains against training efficiency.
>
> ---
>
> ### Q2:
>
> We generate the history summary after the action prediction to ensure that it reflects the most recent action, thereby providing a complete and coherent context for the next step action prediction. Placing the history summary before the action prediction as part of the structured reasoning has two main drawbacks:
>
> (1) The summary at the current step would lack information from the latest action prediction, resulting in an incomplete historical context, which could degrade the performance of subsequent reasoning and decision-making.
>
> (2) The history summary from the previous step is already incorporated as input to the current structured reasoning. Including the current step’s history summary before action prediction would introduce redundant information.
>
> ---

---

> > ### Comment · Area_Chair_ctNx · 2025-08-04
> > **Please respond to the author's rebuttal post**
> >
> > Hi Reviewer 1riz, I see no response letting me know whether or not the rebuttal has changed your opinion. Could you please let me and the authors know by engaging? This process is critical to enabling the (S)ACs to make a decision on this work.
> >
> > --Your AC

---

> > ### Comment · Reviewer_1riz · 2025-08-05
> >
> > Thanks for your detailed explanation and additional experiment results. Most of my concerns are well-explained.
> >
> > I have a bit more to respond to your response in W1. Firstly, it should be possible to integrate the coordinate-based action into WebArena. Considering all other strong baselines that are id-based, it is unfair to compare your method with them. But it is still interesting to see how your method improves Qwen's baseline on WebArena, although the effort might be a bit too much to complete during the rebuttal period. Secondly, the performance of Qwen2-VL-2B on OSWorld might be interesting because your method is based on that model, although it is predictable that the score should be very low.
> >
> > Considering all things, I will maintain my positive score.

---

> ### Author Response · Authors · 2025-08-06
>
> We sincerely appreciate your thoughtful feedback and positive assessment of our work. We will integrate coordinate-based actions in the WebArena evaluation, and include the results in the revised version. We will also report the performance of Qwen2-VL-2B on OSWorld for the completeness of the comparison.

---

### Note · Authors · 2025-08-15

Dear Reviewers and Area Chair,

We would like to express our sincere gratitude for your valuable suggestions and insights. We appreciate all of you for your supportive comments highlighting our work's strengths:

1.  Intuitive framework design validated on GUI tasks (Reviewer 1riz)
2.  State-of-the-art performance across multiple benchmarks (Reviewers Dt5U, W4us)
3.  Clear presentation and rigorous evaluation (Reviewers Dt5U, hw3E)

We also thank the reviewers for their constructive suggestions, which we have addressed:

*   **Expanded evaluation on complex online environments:**
    *   Conducted experiments on Android World, achieving 10.4% step success rate – a 16% improvement over prior 2B SOTA (InfiGUIAgent).
    *   Included OS-World benchmark results (8.7% step success rate), demonstrating robust generalization.
*   **Clarified technical novelty and comparisons:**
    *   Demonstrated that our framework transcends prompt engineering (e.g., 20% performance drop on Mind2Web without structured subtask learning).
    *   Justified history-summary placement (Reviewer 1riz, Q2) and reward design trade-offs (training efficiency vs. marginal gains of +1.7%).
*   **Validated scalability and optimization choices:**
    *   Trained a Qwen2.5-VL-7B parameter model, showing +3% (AITW) and +10% (Mind2Web) gains over SFT baselines.
    *   Explained GRPO’s superiority over PPO (efficiency) and DPO (robustness to overfitting).

We are pleased that our rebuttal has addressed the concerns of Reviewer 1riz, Dt5U and W4us. We are particularly encouraged by:

*   Reviewer 1riz and Dt5U are maintaining a positive score.
*   Reviewer W4us is raising the score to borderline acceptance post-rebuttal.

We will polish the paper in the revised version:

1.  Integrate the experiments discussed in rebuttal (Android World, OS-World, Qwen2.5-VL-7B results) and WebArena adaptation (screen coordinate adaptation).
2.  Expand comparative analysis of history-encoding methods (e.g., UI-Hawk trade-offs) in Related Work.
3.  Correct typos/notation (Fig 1, Eqs 6-7).
4.  Release code, models, and data to ensure full reproducibility.

Finally, we are sincerely grateful that all reviewers recognized the contributions of our work. We are confident our framework will inspire valuable future work in GUI agent research.

---

### Decision · Program_Chairs · 2025-09-17

**Decision:**

Accept (poster)

**Comment:**

The paper claims to provide a novel method for GUI use agents via RL (GRPO). Reviewers agree that the method convincingly works on the benchmarks chosen compared to baselines and design choices are well justified - i.e. correctness is sound. That said, the relative novelty of the method is rather overstated - there appear to only be two things that are new, the history reward during GRPO and the prompt that forms that state representation during RL training. Further, given such specific reward and prompt engineering, it remains unclear how performance would translate to other benchmarks. I will sustain the reviewers' overall suggestion that this paper has enough merits to outweigh such concerns and warrant publication if there is room.